# Effective treatment of mitochondrial myopathy by nicotinamide riboside, a vitamin B3

Nahid A Khan[1], Mari Auranen[1,2,†], Ilse Paetau[1,†], Eija Pirinen[3,4], Liliya Euro[1], Saara Forsström[1], Lotta Pasila[1], Vidya Velagapudi[5], Christopher J Carroll[1], Johan Auwerx[3] & Anu Suomalainen[1,2,6,*]

## Abstract

Nutrient availability is the major regulator of life and reproduction, and a complex cellular signaling network has evolved to adapt organisms to fasting. These sensor pathways monitor cellular energy metabolism, especially mitochondrial ATP production and $NAD^+$/NADH ratio, as major signals for nutritional state. We hypothesized that these signals would be modified by mitochondrial respiratory chain disease, because of inefficient NADH utilization and ATP production. Oral administration of nicotinamide riboside (NR), a vitamin B3 and $NAD^+$ precursor, was previously shown to boost $NAD^+$ levels in mice and to induce mitochondrial biogenesis. Here, we treated mitochondrial myopathy mice with NR. This vitamin effectively delayed early- and late-stage disease progression, by robustly inducing mitochondrial biogenesis in skeletal muscle and brown adipose tissue, preventing mitochondrial ultrastructure abnormalities and mtDNA deletion formation. NR further stimulated mitochondrial unfolded protein response, suggesting its protective role in mitochondrial disease. These results indicate that NR and strategies boosting $NAD^+$ levels are a promising treatment strategy for mitochondrial myopathy.

**Keywords** mitochondrial myopathy; $NAD^+$; nicotinamide riboside; treatment; unfolded protein response
**Subject Categories** Genetics, Gene Therapy & Genetic Disease; Metabolism

See also: **RN Lightowlers & ZMA Chrzanowska-Lightowlers et al** (June 2014)

## Introduction

Mitochondrial disorders, caused by respiratory chain deficiency (RCD), are the most common form of inherited metabolic disorders and manifest with exceptional clinical variability (Ylikallio & Suomalainen, 2011). Despite their progressive and often fatal outcome, no curative treatment is available. Therapy trials for these diseases are mostly hampered by heterogeneity of the disease manifestations and the consequent lack of patient groups with homogenous genetic background and clinical presentations (Suomalainen, 2011). Development of mouse models replicating mitochondrial disease phenotypes has provided a unique opportunity both for therapeutic trials and for detailed studies of molecular pathophysiology of primary mitochondrial dysfunction.

We have previously generated Deletor mice, with adult-onset mitochondrial myopathy (MM) (Tyynismaa et al, 2005). These mice carry a dominant patient mutation in mitochondrial replicative helicase Twinkle, resulting in progressive MM after 12 months of age, with the accumulation of multiple mtDNA deletions in their skeletal muscle and brain, leading to a subtle progressive respiratory chain deficiency (Zeviani et al, 1989; Suomalainen et al, 1992; Tyynismaa et al, 2005). These mice have a normal lifespan and physical performance despite typical morphological changes in MM, cytochrome c oxidase (COX)-negative fibers, and ultrastructural mitochondrial abnormalities, in their muscle. The histological and physiological findings in Deletors mimic closely those of patients with the same mutation (Suomalainen et al, 1992, 1997), making Deletor mice an optimal model for therapy trials.

We found that RCD in Deletor muscles induced a pseudo-starvation response, despite normal food intake, with wide induction of genes with the ATF transcription factor-binding sites in their upstream regulatory regions (Tyynismaa et al, 2010). One of these genes was *FGF21*, encoding a fasting cytokine and leading to increased blood FGF21 concentration and consequent fat mobilization from adipose tissue and liver (Kharitonenkov et al, 2005), without, however, activation of fasting-related beta oxidation or mitochondrial biogenesis. This suggested that RCD induces only some fasting-related pathways, raising the question whether such conflicting signaling, partially promoting catabolism, partially anabolism, could contribute to disease progression. In *C. elegans*, ATFs

1 Molecular Neurology, Research Programs Unit, University of Helsinki, Helsinki, Finland
2 Department of Neurology, Helsinki University Central Hospital, Helsinki, Finland
3 Laboratory of Integrative Systems Physiology, École Polytechnique Fédérale de Lausanne, Lausanne, Switzerland
4 Biotechnology and Molecular Medicine, A.I. Virtanen Institute for Molecular Sciences, Biocenter Kuopio, University of Eastern Finland, Kuopio, Finland
5 Metabolomics Unit, Institute for Molecular Medicine Finland, FIMM, Helsinki, Finland
6 Neuroscience Research Centre, University of Helsinki, Helsinki, Finland
*Corresponding author. Tel: +358 9 4717 1965; Fax: +358 9 4717 1964; E-mail: anu.wartiovaara@helsinki.fi
†These authors contributed equally to the manuscript.

have been linked to mitochondrial unfolded protein response (UPR[mt]) (Nargund et al, 2012), induced by mitochondrial proteostasis, activating ClpP protease (Haynes et al, 2007, 2013), leading to ATF relocalization to nucleus and induction of UPR[mt]-associated gene expression (e.g., mitochondrial heat shock proteins, HSPs). The MM-linked pseudo-starvation response with potential links to UPR[mt] strongly supported the involvement of abnormal nutrient signaling in the disease process.

Nutrient sensors Sirt1 and AMPK monitor tissue $NAD^+$/NADH and AMP/ATP ratios, respectively. If these ratios increased, the sensors induce a metabolic fasting response, boosting mitochondrial biogenesis, fatty acid oxidation, and oxidative ATP production (Imai et al, 2000; Rodgers et al, 2005; Canto et al, 2009, 2012). We hypothesized that RCD reduces NADH utilization, decreasing $NAD^+$/NADH and signaling for high nutrient availability, leaving Sirt1 inactive and attenuating mitochondrial biogenesis. However, RCD also decreases ATP production, increasing AMP/ATP, signaling for low nutrition availability, and leading to a potential conflict in nutrient sensor activation and a partial pseudo-starvation response (Nunnari & Suomalainen, 2012). Previous treatment trials of RC-deficient mice have indicated that activation of mitochondrial biogenesis may be beneficial: High-fat diet (Ahola-Erkkila et al, 2010), PPAR (Wenz et al, 2008; Yatsuga & Suomalainen, 2011) and AMPK agonists (Viscomi et al, 2011) all delayed disease progression. Recently, nicotinamide riboside (NR), a vitamin B3-analogue and $NAD^+$ precursor, was shown to promote oxidative metabolism in mice by increasing $NAD^+$/NADH (Bieganowski & Brenner, 2004; Canto et al, 2012). Furthermore, $NAD^+$ precursors improved mitochondrial function in aging mice to mimic that of young mice, in SIRT1-dependent manner (Gomes et al, 2013). We asked here whether NR could affect the MM pseudo-starvation response and attenuate the disease progression.

# Results

We administered 400 mg/kg/day of NR, or chow diet (CD), for two groups of male Deletor and control mice (Yang et al, 2007; Canto et al, 2012): "pre-manifestation" (12 months at initiation, 16 months at termination), to test whether early NR treatment can prevent MM, and "post-manifestation" (17 months at initiation, 21 months at termination), to test whether NR can affect the progression of an already-manifested disease. The control groups received similar chow diet without NR. The dose and study protocol were exactly as published previously (Canto et al, 2012), because they showed that this dose of NR efficiently increased $NAD^+$ in skeletal muscle.

## NR increased mitochondrial biogenesis in skeletal muscle and ameliorated hallmarks of mitochondrial myopathy

We first analyzed the $NAD^+$ content of Deletor mouse muscle, to clarify whether respiratory chain deficiency affected this pool. Very old Deletor mice, from 23 to 27 months of age, showed a decreasing trend in their skeletal muscle (Fig 1A), whereas younger mice of 17–21 months showed a variation in their $NAD^+$ levels with no clear decrease (Supplementary Fig S1). These results suggest that $NAD^+$ pool was progressively depleted upon mitochondrial myopathy progression.

After 16 weeks of NR treatment, the amount of muscle fibers with decreased cytochrome c oxidase activity (COX-negative), a hallmark of MM and a measure of disease progression, was significantly lower in quadriceps femoris muscle (QF) of both pre- and post-manifestation Deletors compared to the CD-fed Deletors. The post-manifestation Deletors on NR diet showed 1.8-fold less, and pre-manifestation mice 1.5-fold less, COX-negative fibers than Deletors on CD (Fig 1B). The wild-type littermates showed no COX-negative fibers. The overall histochemical COX activity in both oxidative and non-oxidative muscle fibers was increased in all mice receiving NR (Fig 1C), and the amounts of oxidative phosphorylation enzymes having mtDNA-encoded subunits, especially ATPase, were increased (Fig 1D–H). Multiple mtDNA deletions in the Deletor mouse accumulate in the skeletal muscle along with MM progression and are thought to underlie RCD in both mice and men (Zeviani et al, 1989; Suomalainen et al, 1992; Tyynismaa et al, 2005). Consistent with a decrease in COX-negative fibers, both NR-fed Deletor groups had significantly lower mtDNA deletion load in QF than CD-fed Deletors (Fig 1I and J), indicating that their accumulation in Deletor tissues was considerably slowed down. NR treatment increased mtDNA copy number in QF of WT mice, and mtDNA amount also trended upwards in Deletors (Fig 1K). Citrate synthase activity indicated an significantly increased mitochondrial mass in all NR-fed mice (Fig 1L). These results demonstrate that NR induced mitochondrial biogenesis and oxidative ATP production capacity robustly and significantly delayed the progression of MM, irrespective of the disease stage.

## NR prevented the development of mitochondrial ultrastructural abnormalities in mitochondrial myopathy

The electron microscopic examination of skeletal muscle ultrastructure on CD showed swollen mitochondria increased in number, with distorted cristae and subsarcolemmal accumulations, accompanied by myofibrillar degeneration and autophagosomes, especially evident in oxidative, but also in glycolytic fibers (Fig 2A–C), similar to previously reported in MM patients (Suomalainen et al, 1992). On NR diet, the Deletor and WT mice showed a remarkable induction of mitochondrial biogenesis, with an increase in the amount and volume of intermyofibrillar and subsarcolemmal mitochondria (Fig 2D–F and H), with mitochondria-rich pockets around vessels (Fig 2E), similar to those observed in QF of Deletors on high-fat diet (Ahola-Erkkila et al, 2010). Importantly, the mitochondrial ultrastructure of NR-fed Deletors was completely normal, with dense matrices and crista content similar to WT mice on CD (Fig 2F and I), indicating active oxidative metabolism. NR increased further the WT mice matrix and crista density (Fig 2H and I). These findings suggest that NR treatment induces robust mitochondrial biogenesis and prevents the development of ultrastructural abnormalities upon mitochondrial disease.

## NR enhanced lipid oxidation in brown adipose tissue and liver and rescued Deletor BAT phenotype

On NR diet, whole-body $O_2$ consumption and $CO_2$ production of Deletors showed a significant increase upon cold exposure, reaching the maximal gas exchange in 4 h, similar to WT mice. Deletors on CD did not succeed to reach the same level even after 6 h

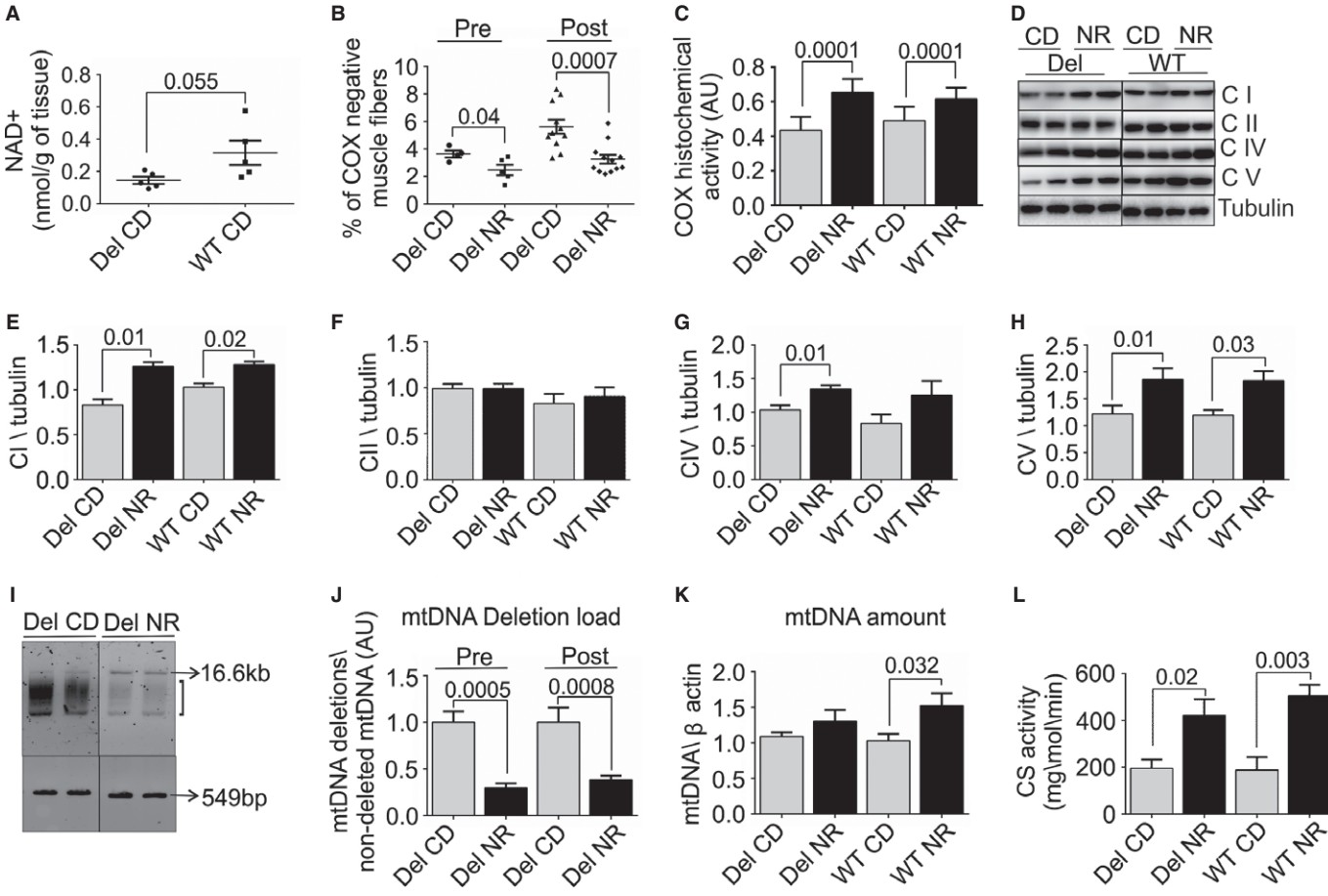

**Figure 1.   Nicotinamide riboside (NR) induces mitochondrial biogenesis and effectively ameliorates morphological and genetic landmarks of mitochondrial myopathy.**

A    Skeletal muscle NAD$^+$ content in 23- to 27-month-old Deletors and WT mice (*n* = 5 in each group).

B    Quantification of muscle fibers showing decreased COX activity in histochemical analysis of frozen sections of *quadriceps femoris* muscle (total fibers *n* = 2000 counted per mouse); diet started before disease manifestation (pre-manifestation, "pre"; Deletor NR *n* = 5; Deletor CD *n* = 4); diet started after disease manifestation (post-manifestation, "post"; Deletor NR *n* = 12; CD *n* = 11).

C    Quantification of the overall histochemical COX activity in muscle. Total intensity of activity in the *quadriceps femoris* muscle measured (*n* = 6 in each group).

D    Mitochondrial respiratory complex protein and ATPase protein amounts in *quadriceps femoris* muscle. Western blot analysis of post-manifestation NR- and CD-fed mouse groups. Representative results shown. CI, respiratory chain complex I, 39-kDa subunit; CII, complex II, 70-kDa subunit; CIV, complex IV, cytochrome c oxidase, COI subunit, 40 kDa; CV, complex V, ATPase, alpha subunit, 55 kDa. Tubulin indicates the loading control.

E–H   Quantification of the respiratory chain complex amounts from Western blots (Fig 1D). (Deletor NR *n* = 8; Deletor CD *n* = 8, WT NR *n* = 6, WT CD *n* = 6).

I    Multiple mtDNA deletions in *quadriceps femoris* muscle. Long-range PCR amplification. 16.6 kb: amplification product from full-length mtDNA; bracket: smear representing deleted mtDNA species; 549 bp, product from 12S rRNA gene in mtDNA, not carrying deletions and representing total mtDNA. Representative image (pre-treatment Deletor NR *n* = 5; Deletor CD *n* = 4; post-manifestation Deletor NR *n* = 12; Deletor CD *n* = 11).

J    Quantification of mtDNA deletion load (H). Pre- and post-manifestation NR- and CD (pre-manifestation; Deletor NR *n* = 5; Deletor CD *n* = 4), (post-manifestation; Deletor NR *n* = 12; CD *n* = 11).

K    mtDNA copy number in *quadriceps femoris* muscle. Quantitative PCR analysis, using beta-actin gene as a nuclear two-copy control gene (post-manifestation Deletor NR *n* = 12; Deletor CD *n* = 11, WT NR *n* = 8, WT CD *n* = 7).

L    Citrate synthase (CS) activity in *quadriceps femoris* muscle (*n* = 5 in each group). Numbers above columns indicate *P*-values. All values are presented as mean ± s.e.m. (Student's *t*-test).

NR, nicotinamide riboside; CD, chow diet; Del, Deletor; WT, wild-type mice; pre, pre-manifestation age group; post, post-manifestation age group; COX, cytochrome c oxidase; CI-CIV respiratory chain complexes I-IV; CV, ATP synthase; AU, arbitrary units.
Source data are available for this figure.

(Fig 3A–D, Supplementary Fig S2A–F), suggesting brown adipose tissue (BAT) dysfunction. Indeed, Deletor BAT showed multiple mtDNA deletions (Supplementary Fig S4A), low lipid amounts (Fig 3E–H), and pale mitochondria with low crista content (Fig 3I–L). NR decreased BAT lipid content and increased mitochondrial

number, size, and matrix density (Fig 3J and L) in both Deletors and WT mice. BAT of Deletors on NR showed 1.7-fold higher mitochondrial crista content than that of Deletors on CD, reaching the level of WT on CD (Fig 3M). Deletor mtDNA amount increased 1.7-fold after NR (Fig 3N). In WT mice, NR also increased mitochondrial crista

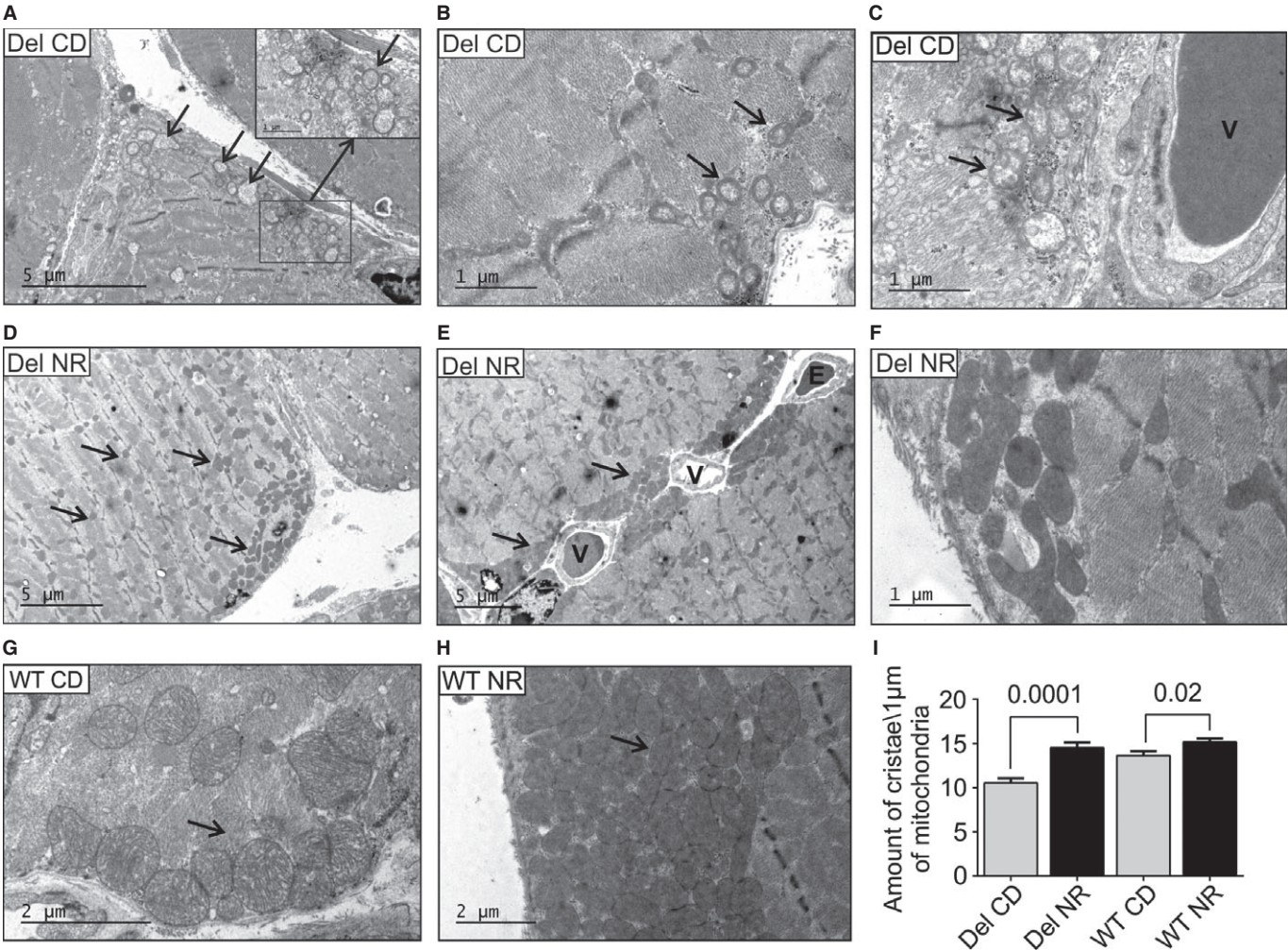

**Figure 2.  NR induces skeletal muscle mitochondrial mass and prevents the development of mitochondrial ultrastructural abnormalities in mitochondrial myopathy.**

A–C  Deletor *quadriceps femoris* muscle ultrastructure on chow diet shows increased amount of swollen mitochondria with concentric crista abnormalities (arrows) especially in subsarcolemmal regions.

D–F  Deletors on NR diet. Mitochondria with normal ultrastructure and dense matrices are prominent subsarcolemmally (D–F), especially around the vessels (E, arrows), and also intramyofibrillar mitochondria show increased volume and matrix density (D, arrows). NR-fed Deletors show high crista density (F).

G  WT mice on CD, showing subsarcolemmal mitochondria (arrow).

H  WT mouse on NR diet with remarkable accumulation of subsarcolemmal mitochondria with dense matrices, and high crista density.

I  Quantification of mitochondrial crista density in Deletors and WT mice on CD or NR diets. In image analysis, a "measuring stick" of 1 μm, placed perpendicular to the crista lamellae, was used to count crista content. A total of 20 μm of mitochondrial length was calculated per each sample electron micrograph. n = 3 in each group.

Data information: Numbers above columns indicate *P*-values. All values are presented as mean ± s.e.m. (Student's *t*-test). Bars in electron micrographs indicate magnification. Del, Deletor; WT, wild-type mice; CD, chow diet; NR, nicotinamide riboside; V, blood vessel; E, erythrocyte.

content and mtDNA amount in BAT (Fig 3K–N). These results indicate that NR induced a robust increase in mitochondrial biogenesis and mitochondrial function of BAT in both WT and MM mice and considerably improved the mitochondrial ultrastructure in Deletors.

The Deletor liver showed low lipid content (Ahola-Erkkila *et al*, 2010; Tyynismaa *et al*, 2010), which was further decreased by NR, similar to WT (Supplementary Fig S3A–D). Liver function tests indicated no manifest liver damage on either diet (Supplementary Fig S3I–J). The mitochondrial ultrastructure was normal in Deletor hepatocytes (Fig 3P and R), which is consistent with the absence of

mtDNA deletions in the livers of Deletors or patients (Suomalainen *et al*, 1992; Tyynismaa *et al*, 2005). This suggests that liver phenotype is secondary to metabolic signals induced by RCD in the skeletal muscle and BAT. NR, however, increased mtDNA copy number in WT (Fig 3O) and changed ultrastructure of rough endoplasmic reticulum, showing close association of the membranes with mitochondria and a suggestive increase in ribosomes on mitochondria in both Deletor and WT (Fig 3Q and S). These findings suggest the induction of metabolic activity in the liver. NR did not clearly affect mtDNA deletion load or copy number in the Deletor brain

(Supplementary Fig S4B and C). Whether NR crosses the blood–brain barrier remains to be confirmed (Canto *et al*, 2012).

### NR enhanced FOXO1 deacetylation, suggesting SIRT1 activation

NR had many beneficial effects on morphological hallmarks of MM and has previously been shown to boost tissue $NAD^+$ levels as well as Sirt1 activity, as measured by FOXO1 deacetylation (Brunet *et al*, 2004; Canto *et al*, 2012). The steady-state $NAD^+$ levels showed an increasing trend in mouse livers and muscle after 16 weeks of NR diet, but these steady-state changes after long-term diet were subtle (Supplementary Fig S1A and B). However, NR significantly decreased the acetylated inactive form of FOXO1 in both Deletor and WT mouse muscle and increased the total amount of FOXO1 in Deletors (Fig 4A–C). These results support the induction of $NAD^+$-stimulated Sirt1 deacetylase function. Activation of fatty acid beta oxidation was suggested by increased mRNA expression levels of long-chain fatty acid transport receptor, CD36, and two enzymes of fatty acid oxidation pathway, ACOX1 and MCAD (Fig 4D–F). These results suggest that NR was capable of activating $NAD^+$-responsive sirtuin signaling and downstream mitochondrial biogenesis.

### RCD induced mitochondrial unfolded protein response and is further enhanced by NR

UPR^mt has been suggested to be beneficial for health and lifespan (Houtkooper *et al*, 2010; Mouchiroud *et al*, 2013) and to overlap with RCD-induced pseudo-starvation response (Tyynismaa *et al*, 2010; Suomalainen *et al*, 2011; Nunnari & Suomalainen, 2012). Of note, we show here that Deletors on CD show the induction of HSP60, HSP70, and ClpP expression, as well as the AARE-regulated FGF21 (Fig 4G and H), indicating that RCD induces UPR^mt through mechanisms that are conserved between worms and mammals (Durieux *et al*, 2011; Houtkooper *et al*, 2013). NR further increased UPR^mt protein amounts. These results indicate that UPR^mt is induced by RCD and suggest that $NAD^+$/NADH contributes to UPR^mt induction.

## Discussion

Mitochondrial myopathy is one of the most common manifestations of adult-onset mitochondrial disorders (Ylikallio & Suomalainen, 2011) and is progressive in nature. Currently, only palliative care can be offered to the patients. We demonstrate here that oral supplementation with NR, a vitamin B3 form and $NAD^+$ precursor, efficiently prevented development and progression of mitochondrial myopathy in mice. NR delayed the development of morphological hallmarks and ultrastructural changes in mitochondria and resulted in a remarkable induction of mitochondrial biogenesis and oxidative metabolism, with an increase in mitochondrial mass, mtDNA, and respiratory chain protein amounts in the muscle. Moreover, NR-fed Deletors showed less mutant mtDNA than their CD-fed littermates, which could be a result of decreased generation of mtDNA deletions or their increased clearance. We show that NR significantly delayed disease progression even in mice that already manifested the disease, which is the time, when medical treatment

would typically be started for patients. These results are remarkable, as they underline the utmost importance of specific vitamin cofactors as modifiers of metabolism in disease, and emphasize the role of nutritional signaling in the pathogenesis of adult-onset mitochondrial disorders.

Mitochondrial biogenesis induction as a treatment strategy was first proposed by Moraes group as their chemical or transgenic induction of PGC1alpha transcription coactivator in severe mitochondrial COX deficiency improved RC function and delayed death of the mice (Wenz *et al*, 2008). These findings were replicated in other models of mitochondrial dysfunction (Viscomi *et al*, 2011; Yatsuga & Suomalainen, 2011). The link to nutritional signaling was underlined by the improvement of mitochondrial pathology after high-fat diet (Ahola-Erkkila *et al*, 2010) and treatment by AMPK agonist AICAR (Viscomi *et al*, 2011) or rapamycin (Johnson *et al*, 2013). In wild-type mice, using an identical treatment protocol as ours, NR was recently shown to increase tissue $NAD^+$ levels and to induce FOXO1 deacetylation, indicating the activation of Sirt1 deacetylase (Canto *et al*, 2012). As Canto *et al*, we found in MM mice signs of sirtuin activation after NR treatment. NR induced FOXO1 deacetylation, which points to Sirt1 activation. The sirtuin-mediated fasting response has potential to activate the downstream target PGC1alpha and mitochondrial biogenesis. This mechanism may underlie the slowed-down disease progression. These findings propose that similar consequences on mitochondrial biogenesis, which have been previously obtained by pharmacological induction of PGC1alpha, can be achieved by vitamin B3 supplementation.

The role of BAT in mitochondrial disease is a little studied field. Patients with a specific mtDNA mutation, m.8344A>G, show multiple symmetric lipomas, which have recently been reported to contain UCP1-positive adipocytes, typical for BAT (Plummer *et al*, 2013). This observation suggested that mitochondrial dysfunction may affect BAT differentiation or function, but the mechanisms involved are unknown. Cold stimulus activates BAT and results in increased oxygen consumption. We found that Deletors responded to cold by increasing their oxygen consumption and $CO_2$ production, but not to similar level as their wild-type littermates, which suggested BAT dysfunction. Indeed, the Deletors showed multiple mtDNA deletions in BAT, and their mitochondrial ultrastructure showed low crista content and matrix density. NR has been previously shown to induce mitochondrial biogenesis in BAT of normal mice (Canto *et al*, 2012). NR effectively cured both the response to cold and mitochondrial ultrastructural abnormalities in Deletors. BAT is an active metabolic organ in mice, but recent years have shown that BAT activity also in humans contributes to lipid metabolism (Virtanen *et al*, 2013). Our results, indicating decreased BAT function in MM, suggest that the tissue may contribute to mitochondrial disease manifestations.

UPR^mt has been suggested to be beneficial for health and lifespan (Houtkooper *et al*, 2010; Mouchiroud *et al*, 2013). We show here that the partial pseudo-starvation response initiated by RCD in skeletal muscle of MM mice shares key components of UPR^mt, previously characterized in detail in *C. elegans* (Haynes *et al*, 2007, 2013; Durieux *et al*, 2011), and not before known to be involved in disease. Therefore, the RCD-induced UPR^mt involves mechanisms that are conserved between worms and mammals. Importantly, NR further enhanced UPR^mt, supporting the previous finding that

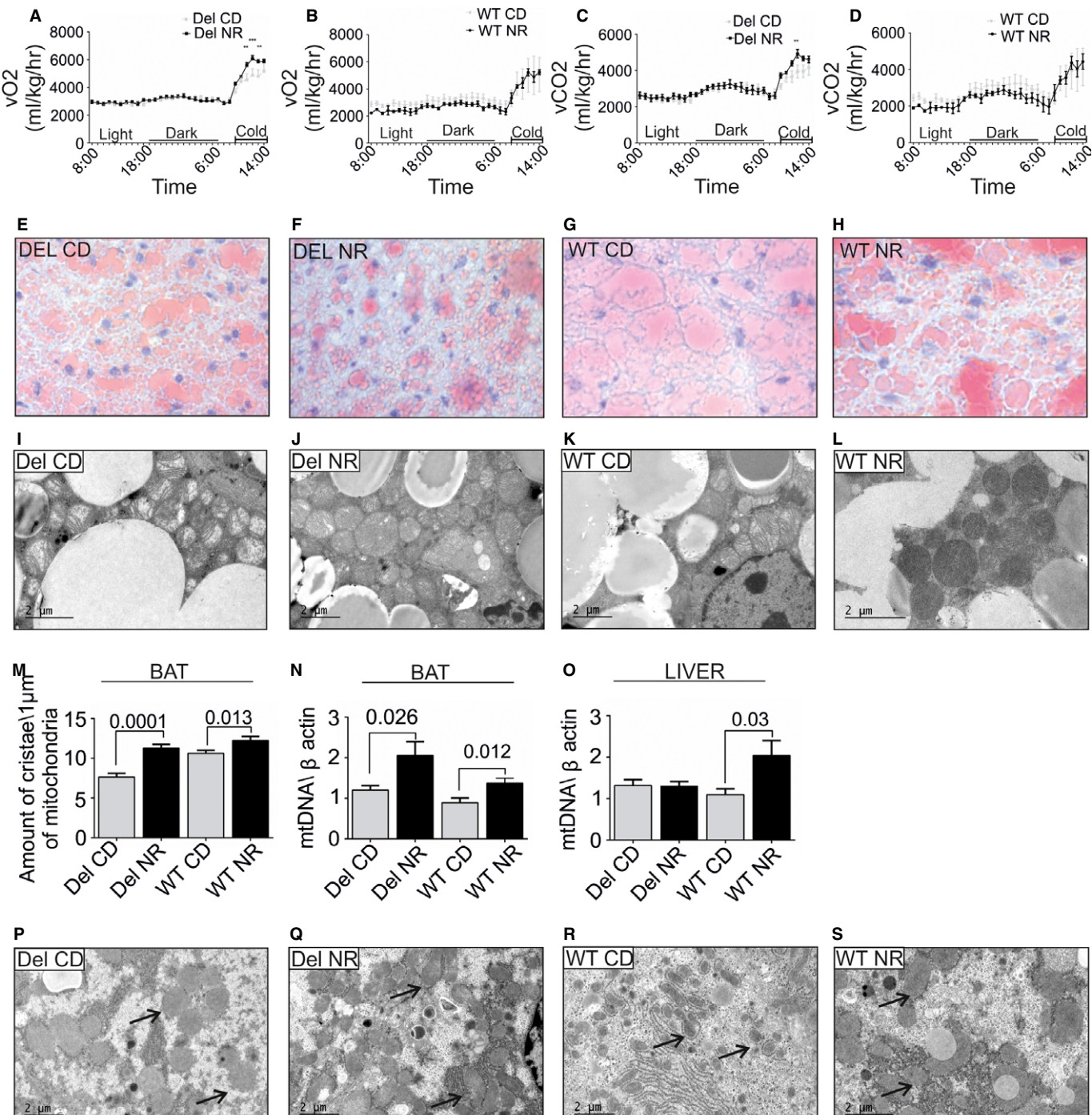

NAD$^+$/NADH contributes to UPR$^{mt}$ induction (Mouchiroud *et al*, 2013), although it is not the sole required signal. Furthermore, improved mitochondrial ultrastructure and function by NR, along with enhanced UPR$^{mt}$, suggests that UPR$^{mt}$ induction is protective for MM.

We have previously described that fasting cytokine FGF21 is a potential biomarker for mitochondrial myopathies (Suomalainen *et al*, 2011), driven by an upstream regulatory AARE element (Tyynismaa *et al*, 2010). Here, we link FGF21 (Kharitonenkov *et al*, 2005) to be part of UPR$^{mt}$. NR further induced UPR$^{mt}$ and FGF21,

while COX-negative fiber amount decreased. Therefore, we propose that FGF21 is a serum biomarker of skeletal muscle UPR$^{mt}$, and part of the pseudo-starvation response induced by RCD, which signals and conveys the local mitochondrial activity and metabolic situation of muscle to the whole organism. FGF21 is a valuable marker for diagnostic use of disorders associated with UPR$^{mt}$. However, upon interventions with NAD$^+$ precursors, or treatment trials enhancing UPR$^{mt}$, also FGF21 would be induced, and in those situations, FGF21 serum concentrations may not reflect muscle treatment response.

**Figure 3.  Decreased brown adipose tissue (BAT) function in Deletors and the effects of nicotinamide riboside treatment for whole-body metabolism, BAT, and liver.**

A, B  Oxygen consumption of post-manifestation NR- and CD-treated Deletor and WT mice. CLAMS® metabolic cage analysis, the final 24-h recording from a total of 72 h. "Light" indicates the hours of lights on, mice inactive. "Dark" indicates the dark hours, mice active. "Cold" indicates cold exposure at 4°C for 6 h. The *P*-values are from left to right: ** = 0.0092, *** = 0.001, ** = 0.0095.

C, D  Carbon dioxide production, as in A–B. **P = 0.013.

E–H  Lipid content of BAT in post-manifestation NR- or CD-fed Deletors and WT mice. Oil Red O lipid staining on frozen sections with hematoxylin–eosin counterstaining. Red indicates lipid pools in BAT. (E) Deletors on CD, (F) Deletors on NR, (G) WT on CD, (H) WT mice on NR diet.

I–L  BAT ultrastructure in NR- or CD-fed Deletors and WT mice. Electron micrographs of Deletor (I, on CD; J on NR) or WT (K on CD, L on NR).

M  BAT, quantification of mitochondrial crista density, as in Fig 2I.

N  BAT, mtDNA copy number, post-manifestation NR- or CD-fed Deletors and WT mice. Analysis by quantitative PCR. (post, *n* = 12 Deletor NR, *n* = 11 Deletor CD, *n* = 8 WT NR, *n* = 7 WT CD).

O  Liver, mtDNA copy number, post-manifestation study groups (*n* = 12 Deletor NR, *n* = 11 Deletor CD, *n* = 8 WT NR, *n* = 7 WT CD).

P–S  Liver ultrastructure in NR- or CD-fed post-manifestation mice. Deletors (P, on CD; Q on NR) and WT mice (R on CD, S on NR). Arrows indicate ribosomes, accumulating in NR-fed mice to close contact with mitochondria. Representative electron micrographs.

Data information: The bars indicate magnification; data shown in A–D (two-way ANOVA); all values are presented as mean ± s.e.m. (Student's *t*-test).

## Materials and Methods

### Animals and study plan

Our data suggest that conflicting nutrient signaling is integral for MM pathogenesis and disease progression. We show that fine tuning of cellular signaling with vitamin cofactors, especially NAD$^+$ precursors, is an intriguing and straightforward therapeutic strategy that should be explored in patients with late-onset mitochondrial myopathy, the most common type of adult-onset mitochondrial disorders.

All animal procedures were performed in accordance with the guidelines by ethical board of State Provincial Office for Animal

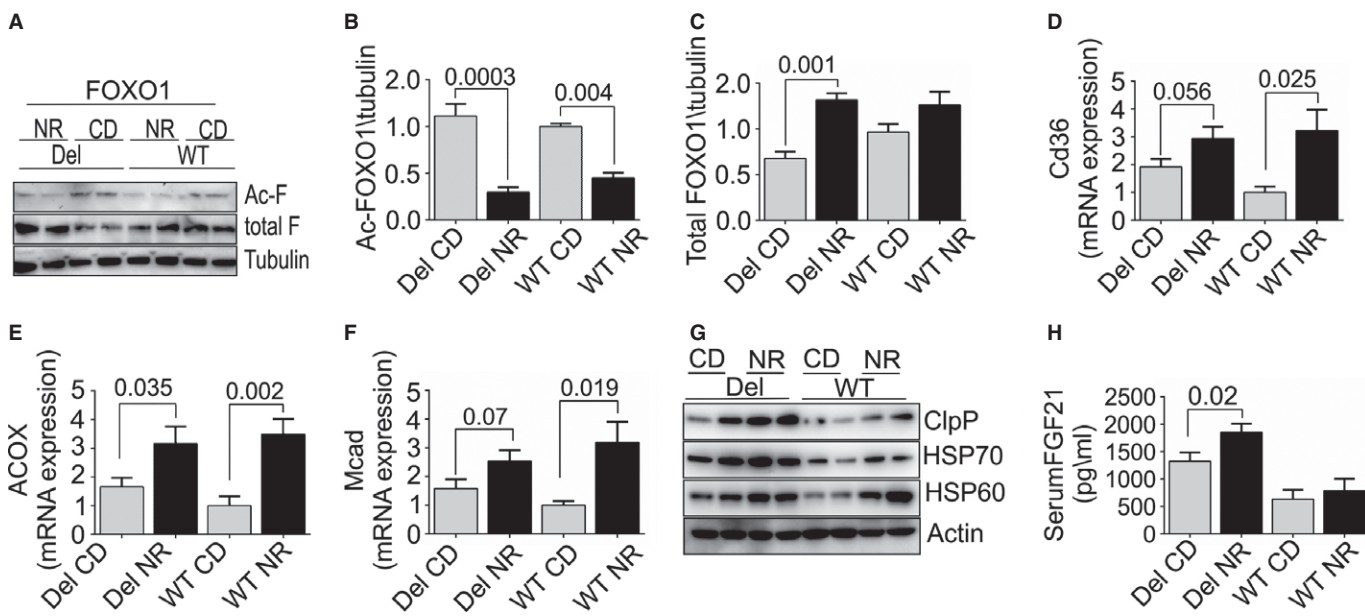

**Figure 4.  Skeletal muscle of Deletor mice have enhanced mitochondrial unfolded protein response (UPR$^{mt}$), expression of fatty acid oxidation enzymes, and deacetylation of FOXO1, a downstream target of Sirt1.**

A  Skeletal muscle, total FOXO1 (total-F), and acetylated FOXO1 (ac-F) protein levels. Western blot analysis, Deletor and WT mice, on CD or NR (*n* = 4 for each group); tubulin as a loading control.

B  Quantification of acetylated FOXO1, from Western blot results (*n* = 4 in each group).

C  Quantification of total FOXO1 (B), from Western blot results (*n* = 4 in each group).

D–F  Muscle mRNA expression of fatty acid transport and oxidation proteins CD36, ACOX1, MCAD (post-manifestation Deletor NR *n* = 12; Deletor CD *n* = 11; WT NR *n* = 8; WT CD *n* = 7; arbitrary units).

G  Muscle UPR$^{mt}$ proteins HSP70, HSP60, ClpP in post-manifestation Deletors, and WT mice on NR- or CD-fed diet. Western blot analysis, beta-actin as a loading control.

H  Serum FGF21 in Deletors and WT mice on CD or NR diet, post-manifestation (Deletor NR *n* = 12; Deletor CD *n* = 11; WT NR *n* = 8; WT CD *n* = 7). ELISA analysis.

Data information: All values are presented as mean ± s.e.m. (Student's *t*-test).
Source data are available for this figure.

     

Experimentation of Finland. The Deletor mouse model was generated in C57BL/6 congenic background and has been previously characterized (Tyynismaa *et al*, 2005); WT mice were littermates from the same congenic mouse strain C57BL/6. Deletor and WT male mice were administered either CD diet (rodent diet with 11% fat, 65% carbohydrate, 24% protein of total calorie content, Altromin Spezialfutter GmbH & Co. KG, Germany) or NR diet (rodent diet as CD, supplemented with 0.24% of NR). NR was custom-synthesized as previously described (Yang *et al*, 2007) and was kindly provided by Amazentis and Merck Research laboratories. The food pellets were manually prepared by baking NR into the powdered food as described previously (Canto *et al*, 2012), and the protocol followed this published protocol in detail. The pellets were stored at −20°C. The mice were housed in standard animal facility, under a 12-h dark/light cycle. They had *ad libitum* access to food and water. The pre-manifestation group consisted of nine Deletors and nine WT mice, and the post-manifestation group of 25 Deletors and 16 WT mice, receiving either NR or CD diet. During the intervention, one mouse died without any notable association with NR treatment (Deletor mouse receiving NR). For $NAD^+$ determination, we used additionally very old Deletor mice from 23 to 27 months of age, on standard diet.

During the intervention, the mice were regularly monitored for weight, food consumption, and physical endurance. Their exercise capability was measured twice by treadmill exercise test (Exer-6M Treadmill, Columbus Instrument) at the start and the end of the diet. The exercise test protocol consisted of the initial running speed of 7 m/s, which was increased every 2 min with 2 m/s and continued until the animal was unable to run or repeatedly fell from the belt at the stimulus site.

### Analysis of whole-body metabolism

Oxygen consumption and carbon dioxide production, as well as spontaneous moving and feeding activities, were recorded by Oxymax Lab Animal Monitoring System (CLAMS; Columbus Instruments, OH, USA). The mice were kept in individual cages inside a CLAMS chamber for 3 days; the first day and night was a non-recording adjustment period followed by a 24-h recording at room temperature (+22°C) and a 6-h cold exposure (+4°C). Measurements were performed for a total of 18 Deletor mice ($n = 8$ on NR and $n = 10$ on CD) and six WT mice ($n = 3$ on NR and $n = 3$ on CD) belonging to the post-manifestation group and for four Deletor mice ($n = 2$ on NR and $n = 2$ on CD) and four WT mice ($n = 2$ on NR and $n = 2$ on CD) from the pre-manifestation group. The results of $O_2$ consumption and $CO_2$ production were used to calculate respiratory exchange rate and analyzed separately from the light (inactive) and dark (active) periods of the day and from the cold exposure.

### Morphologic analysis

Tissue sections were prepared from QF, liver, and BAT. Samples were embedded with OCT Compound Embedding Medium (Tissue-Tek) and snap-frozen in 2-methylbutane in liquid nitrogen. Frozen sections (12 μm) from QF were assayed for *in situ* histochemical COX and succinate dehydrogenase (SDH) activities simultaneously. From QF sections, COX-negative and COX-negative plus SDH-positive and normal fibers were calculated. Approximately 2000 fibers were counted from each mouse sample. The intensity of COX

histochemical activity from QF for both oxidative and non-oxidative fibers was measured with Image J software. Frozen sections (8 μm) from liver and BAT were stained with Oil Red O as previously described (Ahola-Erkkila *et al*, 2010).

For plastic embedding, QF, liver, and BAT samples were fixed in 2.5% glutaraldehyde, treated with 1% osmium tetroxide, dehydrated in ethanol, and embedded in epoxy resin. Semi-thin (1 μm) sections were stained with methyl blue (0.5% w/v) and boric acid (1% w/v). The interesting areas for the ultrastructural analyses were selected by the inspection of light microscopic sections. For transmission electron microscopy, ultrathin (60–90 nm) sections were cut on grids and stained with uranyl acetate and lead citrate and viewed with JEOL 1400 Transmission Electron Microscope. Crista content in both BAT and muscle was calculated as described previously (St-Pierre *et al*, 2003; Canto *et al*, 2012) from electron micrographs, utilizing a 1-μm "intramitochondrial measuring stick," placed perpendicular to cristae.

### CS activity

Skeletal muscle samples were analyzed for citrate synthase activity as described previously (Trounce *et al*, 1996).

### DNA analyses

Total DNA was isolated from snap-frozen QF by standard proteinase K digestion, phenol–chloroform extraction, and ethanol precipitation methods. mtDNA deletion load was determined for each Deletor mouse by semi-quantitative PCR amplification method, as described previously (Ahola-Erkkila *et al*, 2010). mtDNA copy number was analyzed for each mouse by real-time quantitative PCR as described previously (Yatsuga & Suomalainen, 2011), utilizing 12S rRNA gene primers for mtDNA and beta-actin as a control for nuclear two-copy gene.

### Blood analyses

Blood samples were collected with cardiac puncture after mice were sacrificed using carbon monoxide gas. Measurement of alanine transaminase (ALAT) was performed at the Laboratory of the Department of Equine and Small Animal Medicine, the Faculty of Veterinary Medicine, University of Helsinki. From approximately half of the blood volume, plasma was separated by centrifugation of blood at 3,000 *g* for 5 min at +4°C, transferred to 1.5-ml tubes, and maintained at −80°C until analysis. From serum, FGF21 (Quantikine Mouse FGF-21, R&D Systems) was quantified by ELISA assay. Microplates were read by Spectra-Max 190 Absorbance Microplate Reader (Molecular Devices, Inc., USA) and analyzed by SoftMax Pro (Molecular Devices, Inc.).

### Gene expression analyses

Total cellular RNA from frozen muscle tissue and liver samples were extracted in TRIzol reagent (Invitrogen) and homogenized with Fast-Prep w-24 Lysing Matrix D (MP Biomedical) and Precellys w-24 (Bertin Technologies) for 15 s. Two micrograms of total RNA was used to generate cDNA using Maxima first-strand cDNA synthesis kit (K1641; Thermo Scientific). Specificity of amplification was tested by

PCR and agarose gel electrophoresis. Quantitative real-time PCR amplification of cDNA generated from mouse tissues was performed with DyNAmo Flash SYBR Green qPCR kit (Finnzymes) on CFX96 Touch qPCR system (Bio-Rad) and quantified on exponential phase of PCR. Primer sets are available from authors on request.

### NAD$^+$ determination

HPLC analysis: For NAD$^+$ extraction, 500 μl of 1.66 M HClO$_4$ was added to the 4–10 mg pieces of frozen tissues and homogenized in CKMix tubes (Precellys), thrice, for 20 s each, at 4,000 rpm of homogenizing speed in Precellys homogenizer with 2-min breaks on ice. Tissue homogenates were centrifuged at 20,000 $g$ for 10 min, +4°C. 117 μl of 3M K$_2$CO$_3$ was added to 350 μl of supernatants, exothermic reaction stopped, and the samples were vortexed and centrifuged at 14,000 rpm for 5 min at +4°C. Supernatants containing NAD$^+$ were frozen at −80°C. NAD$^+$ extracts were thawed and centrifuged at 14,000 rpm for 5 min at +4°C and quantified on HPLC column Poroshell 120 EC-C18, with dimensions 3.0 × 50 mm and particle size 2.7 μm (Agilent) using ÄktaPurifier UPC10 system (GE Healthcare). The column was equilibrated with 9.6 mM KH$_2$PO$_4$, 154 mM K$_2$HPO$_4$, pH 7.02. Flow rate was 0.6 ml/min, and detection was at 254 nm. Fifty microliters of the NAD$^+$ extract was injected into the column and eluted isocratically with the same mobile phase. Retention volume for NAD$^+$ varied between 0.68 and 0.74 ml. NAD$^+$ content in the extracts was calculated based on the NAD$^+$ standard curve for concentrations from 0.5 to 10 μM and normalized on mass of tissue.

Mass spectrometry analysis: Polar metabolites were extracted from mouse muscle samples, separated using Waters Acquity ultra performance liquid chromatography, and analyzed using triple quadrupole mass spectrometry. For method details, see Supplementary information.

### Western Blotting

Whole-tissue lysates were prepared from a piece of QF homogenized in ice-cold PBS using a Precellys homogenizer and lysed in RIPA buffer with protein inhibitor (Complete Mini, Roche), followed by incubation on ice for 20 min. Protein concentration was measured using the Bradford method (Protein Assay, Bio-Rad). Ten micrograms of protein was loaded per well in SDS–PAGE and transferred to Immobilon-FL transfer membrane (Millipore) using semi-dry transfer unit (Hoefer TE70, Amersham Biosciences). The membranes were blocked in 5% milk in TBS 0.1% Tween. Antibody detection reaction was developed by enhanced chemiluminescence (Bio-Rad). The anti-FOXO1 antibody was from Cell Signaling, and the beta-tubulin antibody was from Sigma; HSP70 and mitochondrial respiratory complex antibodies were from Abcam, against CI, respiratory chain complex I, 39-kDa subunit; CII, complex II, 70-kDa subunit; CIV, complex IV, cytochrome c oxidase, COI subunit, 40 kDa; CV, complex V, ATPase, alpha subunit, 55 kDa, ClpP antibody from Proteintech, and antibodies for HSP60, acetylated-FOXO1 (sc-43497), and beta-actin were from Santa-Cruz.

### Statistics

All results in the figures are expressed as mean ± s.e.m., unless otherwise specified. One-way ANOVA was used for mice and food

## The paper explained

### Problem
Mitochondrial disorders are the most common cause of inherited metabolic disease of adults and children, with no means of cure. Physiological studies utilizing mouse models have suggested that energy metabolic defect in adult muscle is interpreted by some cellular pathways as a state of starvation, despite normal nutrition, leading to "pseudo-starvation" response. One of the cellular mediators for this response is the ratio of oxidized (NAD$^+$) to reduced (NADH) nicotinamide adenine dinucleotide. Vitamin B3 is a direct precursor of NAD$^+$. We asked here whether increasing cellular NAD$^+$ levels by a vitamin B3 form, nicotinamide riboside (NR), could alleviate pathological changes upon mitochondrial myopathy (MM) or delay disease progression.

### Results
We report that per-oral NR, a vitamin B3 form and NAD$^+$ precursor, effectively delayed mouse MM progression. NR robustly induced mitochondrial mass and function and cured structural abnormalities of mitochondria, as well as delayed accumulation of mitochondrial DNA mutations. We show here that MM pseudo-starvation response is linked with a protective stress response, mitochondrial unfolded protein response (UPRmt), with induction of fasting cytokine FGF21. NR further enhanced UPRmt, supporting a protective role of UPRmt upon MM.

### Impact
We show that the conflicting nutrient signaling is an integral part of MM and that fine tuning of the cellular signaling with oral administration of vitamin cofactors, such as NR, is highly beneficial and attenuating the disease progression of disease phenotype. These results indicate that vitamin cofactors modify metabolism and that treatment strategies increasing NAD$^+$ should be explored in the patients.

weight, blood and serum tests, and CLAMS data. Other studies were evaluated using unpaired Student's *t*-test. Statistical analyses were performed with PRISM 6.0 (GraphPad software).

**Supplementary information** for this article is available online: http://embomolmed.embopress.org

## Acknowledgements
The authors wish to thank technical contributions of Markus Innilä, Anu Harju, and Tuula Manninen. AS is supported by grants from the European Research Council, Jane and Aatos Erkko Foundation, Sigrid Jusélius Foundation, Academy of Finland, and University of Helsinki. JA is the Nestlé Chair in Energy Metabolism. JA is supported by grants of the Ecole Polytechnique Fédérale de Lausanne, the ERC, the NIH (R01HL106511-01A and R01AG043930), the Velux Stiftung, and the Swiss National Science Foundation. EP is funded by the Academy of Finland, the Saastamoinen Foundation, the Finnish Cultural Foundation, and the Finnish Diabetes Foundation. CJC is supported by a Finnish Academy post-doctoral fellowship.

## Author contributions
NAK, MA, IP, CJC, and AS designed and performed experiments, analyzed results, and contributed to writing of manuscript. EP, LE, SF, LP, and VV performed experiments and interpreted the results. JA and AS designed the study and contributed to the writing of manuscript.

## Conflict of interest

The authors declare that they have no conflict of interest.

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
