## [Review Process File · EMBO Molecular Medicine]

Effective Treatment of Mitochondrial Myopathy by Nicotinamide Riboside, a Vitamin-B3

Nahid A Khan, Mari Auranen, Ilse Paetau, Eija Pirinen, Liliya Euro, Saara Forsström, Lotta Pasila, Vidya Velagapudi, Christopher J Carroll, Johan Auwerx and Anu Suomalainen

Corresponding author: Anu Suomalainen, University of Helsinki

Review timeline:	Submission date:	06 February 2014
	Editorial Decision:	27 February 2014
	Revision received:	06 March 2014
	Editorial Decision:	10 March 2014
	Revision received:	14 March 2014
	Accepted:	17 March 2014

Transaction Report:

Editor: Roberto Buccione

1st Editorial Decision

27 February 2014

Thank you for the submission of your manuscript to EMBO Molecular Medicine. We have now received reports from the three Reviewers whom we asked to evaluate your manuscript

You will see that all three Reviewers are generally supportive of your work and underline its considerable potential interest. They do however express a few nicely complementary concerns that require your attention and action before publication of the manuscript can be further considered. I will not go into much detail as the comments are to the point and quite clear.

Reviewer 1 has a few concerns regarding presentation of the data, which need to be addressed; I would especially emphasize the issues with Fig.4 that appear to be shared by all three Reviewers.

Reviewer 2 also lists a number of point that require further reflection and discussion, including some imprecise reference to literature data on your part. Again, s/he is not convinced at all about the quality and significance of Fig. 4.

Reviewer 3, while also recognizing the interest of your work, is slightly more critical (including on Fig. 4). S/he feels that the data linking the effect of NR through NAD⁺ is not especially compelling and I tend to agree. This Reviewer would also like to see additional information from other tissues. All considered, the Reviewers are not arguing that the effect of NR is not striking but I would argue that the molecular mechanisms for this effect are not as clear as we would like them to be.

However, considered that a potentially simple therapeutic for use in treating mitochondrial disease, clearly an unmet need, is a relevant issue and that this aspect of your work does not appear to be

under question, I would ask you to tone down your claims on the mechanism of action of NR or, ideally, strengthen them experimentally. Furthermore, the issues raised by Reviewer 3 need to be discussed adequately. If you have data on other tissues, this would also increase interest and impact of your work but I would not consider this vital at this stage.

We would be thus pleased to consider a suitably revised version provided all the other issues are dealt with carefully (including on Fig. 4). I am prepared to make an Editorial decision, provided you adequately address all concerns as discussed above.

Please note that it is EMBO Molecular Medicine policy to allow a single round of revision only and that, therefore, acceptance or rejection of the manuscript will depend on the completeness of your responses included in the next, final version of the manuscript.

I look forward to receiving your revised manuscript as soon as possible.

***** Reviewer's comments *****

Referee #1 (Remarks):

Figure 1C.-The antibodies used for immunodetection should be indicate. The extrapolation of the SDS WB of one protein of each complex as a valuable indication of the amount of fully assembled complex can generate wrong results. In addition, the blot shown do not reflect the quantitative estimation in the following panels particularly for CI vs tubulin.

Figure 2 and 3.- I was unable to find the information of the number of elements qualified in the electronmicrographs to reach the quantitative data

Figure 4A.- The effect on the NR diet is suggestive of increasing the NAD⁺ levels but it is not significant. Would be significant the increase in the NAD⁺/NADH ratio?

Figure 4B.- The blot of Ac.FOXO1 lack of sufficient quality to be shown, if it cannot be improved is better to remove it.

Referee #2 (Remarks):

Thank you for sending me this manuscript by Khan et al. It is a very thorough and straightforward analysis to show that treatment of a mouse model of a mtDNA deletion-associated mitochondrial disease with nicotinamide riboside is highly effective. This is of particular importance as there is currently no cure or even effective treatment for the majority of defects associated with mitochondrial disease. The data is on the whole, very convincing and expts are professionally performed. There is little to criticise, but I do have some questions and comments, below.

1. Why was the figure of 400 mg/kg/day used for the NR ? Please explain or reference.
2. It is interesting in figure 1 that there is a greater increase in CS activity than for any of the OXPHOS components. Perhaps the authors could comment ?
3. The biggest surprise to me is the relative loss of the depleted mtDNA in the NR treated mitos. What causes this ? Accepting there is an NR-promoted increase in mito biogenesis and mtDNA copy number, why is the delete molecule lost ? Is there a problem with replication of the delete molecules ? Why, then, do they accumulate with age, or perhaps they do not ? Is there some effective segregation of the delete molecules into organelles that are degraded ? Of course, this is not central to this piece of work, but its really interesting. There is no a priori reason why mito biogenesis would lead to the loss of the delete molecule.
4. Fig 3Q, S there is a claim that there is an apparent increase in mitochondrially-associated ribosomes. I would be very careful of making this statement without some kind of labelling support

with relevant antibodies.

5. My one quibble with the data is the claim that on NR treatment there is evidence that acetylated FOXO1 decreases (Fig 4B, upper panel). Perhaps it is just my copy, but this is not convincing.

Otherwise, everything appears pretty solid.

6. Finally, there is the claim that the role of BAT and mito disease is completely unstudied. There is published evidence that BAT-derived lipomas in MERRF are stacked with mutated mtDNA.

7. My final comment is about NR. There have been so many anecdotal reports of efficacy of vitamin cocktails for the treatment of mito disease over the years. These expts with NR have tried to bring some impressive molecular cell biology to support the claims that at least for NR there is some well founded and supported rationale for this particular treatment.

Referee #3 (Comments on Novelty/Model System):

This study does not provide enough evidence linking respiratory chain disease to decreased NAD⁺ levels in muscle, that is the main hypothesis of the study. The most likely explanation is that the mouse model used has a very subtle biochemical phenotype. In fact, NAD⁺ levels in the tissues examined (apparently only the liver) are not significantly different from those of WT mice. This is at most an indirect evidence of their hypothesis. Analysis of a further mouse model with a clearer phenotype would greatly improve the paper.

Referee #3 (Remarks):

The Authors have previously shown the presence of a pseudo-starvation response in their model, the Deletor mice, implying involvement of abnormal nutrient signalling in this disease model. In the present work, they hypothesize that respiratory chain dysfunction, by reducing NADH utilization and thus the NAD⁺: NADH⁺ ratio attenuates mitochondrial biogenesis mediated by the nutrient sensor Sirt1.

Since induction of mitochondrial biogenesis could be beneficial in the treatment of respiratory chain disease, the authors modify the supposed NAD/NADH⁺ imbalance due to an inefficient respiratory chain function in their Deletor model, by administering NAD⁺ to the animals. In fact NAD⁺ has been previously shown by Cantú et al in 2012 to promote mitochondrial biogenesis and oxidative metabolism in ageing mice by increasing the NAD⁺/NADH ratio.

The results of the study nicely confirm the previous observations by Cantú et al. Furthermore they provide evidence for a positive effect of NR on the mitochondrial unfolded protein response. Based on their observations, the Authors propose that chronic NR supplementation, by inducing mitochondrial biogenesis, slows down the progression of mitochondrial myopathy in the Deletor mice, and suggest this therapeutic strategy for patients with late-onset mitochondrial myopathy.

1) My major concern regards the demonstration of a clear link between RCD and decreased NAD⁺ in their mice. In other words: while the Authors clearly demonstrate that chronic NR administration increases mitochondrial biogenesis, respiratory chain complexes proteins, and histochemical COX activity to the same extent in the Del and in control mice, they do not provide enough evidence linking lower levels of NAD⁺ to RCD in the skeletal muscle of Del mice, before NR treatment. In fact, they provide information about NAD⁺ levels only in liver (as stated in the text, results section) and these are not significantly different from controls. In addition, according to the legend of figure 4A, where NAD⁺ values are reported, evaluations were performed on liver and muscle. If this was the case, they are not kept distinct in the graph, which is confusing. Moreover, since there is no clear evidence of a defective respiratory chain in the Del mouse, that only shows morphologic (histochemical and ultrastructural) features consistent with mitochondrial myopathy, we are aware that could be hard to demonstrate a NAD⁺ reduction.

To confirm the Author's hypothesis, it would be important to use a mouse model with a more clear phenotype of mitochondrial myopathy, possibly also with evidence of impaired exercise capacity.

2) A second issue regards the analysis of other tissues besides skeletal muscle: it is always good to provide additional data that may help in understanding the mechanisms at play. However,

sometimes it is difficult to understand from the text and the figures which results were obtained in which tissue, as is the case for the paragraph on steady state NAD⁺ amount and FOXO deacetylation and the relative figures 4 A-F. The results are mixed up in the text and figures and should be kept separated, since the main scope of this work is to demonstrate that NAD⁺ is effective in slowing down the progression of mitochondrial myopathy.

3) A third issue concerns the finding of significantly lower mutation load in the muscle of Del mice receiving NAD⁺. Could the Authors comment on the link between the induction of mitochondrial biogenesis by NAD⁺ and the decreased amount of mtDNA deletion in muscle?

1st Revision - authors' response

06 March 2014

Detailed replies to the Reviewers' criticism

The authors would like to thank the Editor and the Reviewers of their constructive and positive comments. We are happy to hear that the Editor and the Reviewers share our view on the main conclusions of the manuscript, the remarkable effect of NR in mitochondrial myopathy.

Please find below our detailed responses to the Editorial and Reviewers' comments, including new/improved data on NAD⁺ levels and FOXO1 acetylation.

Editor: Reviewer 1 has a few concerns regarding presentation of the data, which need to be addressed; I would especially emphasize the issues with Fig.4 that appear to be shared by all three Reviewers.

Authors: Because of the uniform view of the Editor and the Reviewers of the original Fig 4, concerning the FOXO1 acetylation results and NAD⁺ levels, we have now performed extensive new experimental setup, to improve the quality of the result.

- We provide a new blot of acetylated FOXO1 and total foxo, and the results are stronger, but the conclusions are as in the original paper: FOXO1 acetylation is reduced by NR diet, both in WT and in Deletors, supporting Sirt1 activation. The new blot is now presented in Fig 4A, and the quantification total-FOXO and ac-FOXO signals are presented in Fig 4B and C.
- We succeeded to analyze NAD⁺ levels in the muscle of Deletors, by a mass spectrometric analysis, whereas we had previously data only from the liver in the original manuscript. We noticed that NAD⁺ levels in the 21-month old Deletors showed some trend of being reduced in normal diet, but when we analyzed considerably older mice, up to 27 month-old Deletors and controls, Deletor muscle appeared to have reduced NAD⁺ content, reaching border-line significance, p=0.059. These results support our initial hypothesis of lowered NAD⁺ levels in the affected tissue of Deletors, and the rationale of using NAD⁺ precursors for these animals. We have inserted the old-Deletor results as figure 1A, and shifted the other NAD⁺ measurement results to Supplementary Figure 1.

Editor: Reviewer 2 also lists a number of point that require further reflection and discussion, including some imprecise reference to literature data on your part. Again, s/he is not convinced at

all about the quality and significance of Fig. 4.

Authors: Reviewer 2 only wished to see only one additional recent reference, which has been added.

Editor: Reviewer 3, while also recognizing the interest of your work, is slightly more critical (including on Fig. 4). S/he feels that the data linking the effect of NR through NAD⁺ is not especially compelling and I tend to agree. This Reviewer would also like to see additional information from other tissues. All considered, the Reviewers are not arguing that the effect of NR is not striking but I would argue that the molecular mechanisms for this effect are not as clear as we would like them to be.

Authors: NR effect on mitochondrial biogenesis has been demonstrated in detail on wild type mice (previous work by Johan Auwerx's, a co-author in the current manuscript in Canto et al. Cell Metab 2013). They showed the molecular consequences of NR, nicotinic acid and nicotinamide mononucleotide on NAD-metabolism in cultured cells and also in different tissues of wild type mice. They also showed the increase of NAD⁺ in tissues and activation of sirtuins after NR. They stated: "All compounds increased NAD⁺ levels in liver, but only NR and NA increased the muscle NAD⁺ content". The clearest changes were seen in the liver. Because the evidence of NR-linked NAD⁺ pool increase was strong, we chose the exactly same dose and protocol of NR for our treatment trial. We focused on the effect on the disease, but we also replicated some key findings of Canto et al., including the NAD⁺ content in the liver and muscle, as well as FOXO-acetylation, indicating Sirt1 activation. The most remarkable effect, however, is in mitochondrial biogenesis. In our opinion, the previous literature on WT mice, with our current results on mitochondrial disease mice indicate clearly that NR reached the tissues of our mice, and affected the NAD⁺ pool. Furthermore, as explained above, we have now inserted NAD⁺ measurements from old Deletor mice, with decreased NAD⁺ levels in their skeletal muscle in normal diet, answering the major comment of the Reviewer 3.

Editor: However, considered that a potentially simple therapeutic for use in treating mitochondrial disease, clearly an unmet need, is a relevant issue and that this aspect of your work does not appear to be under question, I would ask you to tone down your claims on the mechanism of action of NR or, ideally, strengthen them experimentally. Furthermore, the issues raised by Reviewer 3 need to be discussed adequately. If you have data on other tissues, this would also increase interest and impact of your work but I would not consider this vital at this stage.

Au: Thank you, we believe we have strengthened the specific mechanistic result concerning the FOXO1 deacetylation and NAD⁺ levels, and otherwise we have scanned the paper not to state more than the results justify. For example, we have omitted the only sentence stating that NAD⁺ levels were increased after NR diet, but only speak of a trend.

Referee #1 (Remarks):

Reviewer: Figure 1C.-The antibodies used for immunodetection should be indicate.

Authors: Thank you for pointing out an obvious issue. The information has been added to the methods and to the figure 1C.

Reviewer: The extrapolation of the SDS WB of one protein of each complex as a valuable indication

of the amount of fully assembled complex can generate wrong results. In addition, the blot shown do not reflect the quantitative estimation in the following panels particularly for CI vs tubulin.

Authors: In tissues, according to our own experience, typically a lack of a specific subunit quite well correlates with the amount of the full complex, instead of just the subunit that is detected in SDS-PAGE. We chose to present representative images, but the quantifications were done from a total of 8 MM mice and 6 of wt-mice per each study group. We do agree with the reviewer that the change was not apparent in the figure, and have now revised the figure 1D concerning CI results.

Reviewer: Figure 2 and 3.- I was unable to find the information of the number of elements qualified in the electronmicrographs to reach the quantitative data.

Authors. As described in the Methods, we utilized a 1 μm “measuring stick”, positioned perpendicular to the cristae of the mitochondria. We have now added to the figure legend and the Methods that 20 μm of mitochondrial matrix per animal (n=3 mice per study group) was analyzed.

Reviewer: Figure 4A.- The effect on the NR diet is suggestive of increasing the NAD⁺ levels but it is not significant. Would be significant the increase in the NAD⁺/NADH ratio?

Authors: This is a good question, but we have unfortunately not been able to measure NADH reliably from tissues. Measurement of steady state NAD⁺ and NADH is challenging from tissues, and is typically done in cultured cells or young mice. We tested many commercial kits for both NAD⁺ and NADH measurement, and found none reliable, and then set up NAD⁺ HPLC analysis, and now also ultra performance liquid chromatography and triple quadrupole mass spectrometry. We show here the in the liver, NAD⁺ content shows a clear increasing trend in both WT and Deletor mice after NR treatment. Our new data now shows additionally, that in old Deletor mice in CD, muscle shows lowered NAD⁺ levels.

Reviewer: Figure 4B.- The blot of Ac.FOXO1 lack of sufficient quality to be shown, if it cannot be improved is better to remove it.

Authors: We have been now able to improve this result, and a new blot has been inserted. The conclusions were similar as originally, but the results are more obvious.

Referee #2 (Remarks):

Thank you for sending me this manuscript by Khan et al. It is a very thorough and straightforward analysis to show that treatment of a mouse model of a mtDNA deletion-associated mitochondrial disease with nicotinamide riboside is highly effective. This is of particular importance as there is currently no cure or even effective treatment for the majority of defects associated with mitochondrial disease. The data is on the whole, very convincing and expts are professionally performed.

Authors: We thank the Reviewer on these highly positive notes.

Rev: 1. Why was the figure of 400 mg/kg/day used for the NR ? Please explain or reference.

Authors: This same dose had been used in a previous study by Canto et al, Cell Metab, 2012, for wild type mice. The authors showed that this dose increased considerably NAD⁺ levels in mouse tissues and induced mitochondrial biogenesis. Therefore we applied the same dose and protocol for mitochondrial myopathy mice. This information was stated and referenced in the original paper, but to clarify it further, we added a note in results as well as in methods.

Reviewer: It is interesting in figure 1 that there is a greater increase in CS activity than for any of the OXPHOS components. Perhaps the authors could comment ?

Authors: Both OXPHOS protein amounts and CS activity show an increase after NR diet. However, it is not straightforward to compare protein amounts and activity, and therefore we only conclude that both are increased and support induction of mitochondrial biogenesis.

Reviewer: The biggest surprise to me is the relative loss of the depleted mtDNA in the NR treated mitos. What causes this ? Accepting there is an NR-promoted increase in mito biogenesis and mtDNA copy number, why is the delete molecule lost ? Is there a problem with replication of the delete molecules ? Why, then, do they accumulate with age, or perhaps they do not ? Is there some effective segregation of the delete molecules into organelles that are degraded ? Of course, this is not central to this piece of work, but its really interesting. There is no a priori reason why mito biogenesis would lead to the loss of the delete molecule.

Authors: The authors agree with the reviewer that this is very interesting. However, as the Deletor tissues were analyzed only after both NR vs CD groups were sacrificed, we do not know of the dynamics of the mtDNA deletions in vivo. The options are either that mtDNA mutagenesis occurred in a slower pace, or that the generated deletions were cleared out in an enhanced pace. To get understanding on this subject, labeling experiments on mtDNA and determination of its half-life would be required, as well as crossing of MM mice with autophagy marker-mice. These experiments are obvious next steps, but as they also are long-term studies, taking over 2 years, they are not in the scope of this article. We have now added to the Discussion: “Moreover, NR-fed Deletors showed less mutant mtDNA than their CD-fed littermates, which could be a result of decreased generation of mtDNA deletions, or their increased clearance. “

Reviewer: Fig 3Q, S there is a claim that there is an apparent increase in mitochondrially-associated ribosomes. I would be very careful of making this statement without some kind of labeling support with relevant antibodies.

Authors: We do agree on this comment, and have now revised the conclusions to be even more careful. We have omitted discussion of protein synthesis induction, stating now only (p.5): “These findings suggest induction of metabolic activity in the liver.” We still mention the apparent increase of mitochondrial-ER association, as the figures were representative images of a series of electron micrographs, and as the other reviewers did not criticize this – the observation may be important, but require detailed attention.

Reviewer. My one quibble with the data is the claim that on NR treatment there is evidence that acetylated FOXO1 decreases (Fig 4B, upper panel). Perhaps it is just my copy, but this is not convincing. Otherwise, everything appears pretty solid.

Authors: We agree, and have now been able to verify the result, supporting the original conclusion. We have inserted a new Western figure and quantification, clearly indicating decreased FOXO1

acetylation after NR diet in skeletal muscle, which supports well the findings of Canto et al in cultured cells.

Reviewer: Finally, there is the claim that the role of BAT and mito disease is completely unstudied. There is published evidence that BAT-derived lipomas in MERRF are stacked with mutated mtDNA.

Authors: We thank the reviewer, as we had not noticed the recent publication describing MERRF patients with UCP1-positive lipomas – however, the mechanisms of why or how this specific mtDNA mutation generates BAT-like lipomas are completely open. We now modified the sentence to state “little studied” instead of “completely unstudied” field. We have also now added the reference (Plummer et al. Mitochondrion 2013), and a sentence mentioning it, to the manuscript discussion, p. 7.

Reviewer: My final comment is about NR. There have been so many anecdotal reports of efficacy of vitamin cocktails for the treatment of mito disease over the years. These expts with NR have tried to bring some impressive molecular cell biology to support the claims that at least for NR there is some well founded and supported rationale for this particular treatment.

Authors: We fully agree with the Reviewer. It is also notable, that despite vitamin cocktails being used, NAD⁺ precursors have not been part of these cocktails. We do strongly feel that controlled clinical trials in patients with B3 are warranted after these findings.

Referee #3 (Comments on Novelty/Model System):

Reviewer: This study does not provide enough evidence linking respiratory chain disease to decreased NAD⁺ levels in muscle, that is the main hypothesis of the study. The most likely explanation is that the mouse model used has a very subtle biochemical phenotype. In fact, NAD⁺ levels in the tissues examined (apparently only the liver) are not significantly different from those of WT mice. This is at most an indirect evidence of their hypothesis.

Authors: We should point out that liver mitochondria do not show mtDNA deletions in the Deletor, and liver is not a primary affected organ in Deletor mice – therefore we would not expect to see big changes in CD-fed Deletors vs CD-WT in the pre-treatment situation (Tynismaa et al. PNAS 2005 and Hum Mol Genet 2010). We did liver measurements, as this tissue works well in our hand in NAD⁺ analysis. Measurement of steady state NAD⁺ and NADH is challenging from tissues, and is typically done in cultured cells or young mice. We had set up HPLC analysis for NAD⁺ content, and now also ultra performance liquid chromatography and triple quadrupole mass spectrometry. We show here new data that NAD⁺ content in old mice, 23-27 month old Deletors, is decreased in the skeletal muscle, supporting our hypothesis that NAD⁺ may be low in mitochondrial disease. In younger mice – 21-olds, which was the eldest time point in NR treatment groups – we see more variation in both control and NR-treated groups in skeletal muscle. However, liver shows a trend in increasing NAD⁺ content both in WT and Deletor mice after NR treatment.

We felt it important to show these results, even if borderline significant, as they supported well the findings of Canto et al, despite our results being from considerably older mice than theirs. We have, however, demonstrated remarkable changes after NR diet on induction of mitochondrial biogenesis in the skeletal muscle and brown adipose tissue, as well as on the whole-body oxygen consumption and CO₂ production. These results alone are a strong indication that the NR compound did reach the tissues of interest - there is no doubt about that – and that it induced mitochondrial biogenesis. Now we also show that NAD⁺ seemed reduced in Deletor muscle without treatment, which was the major concern of this reviewer. This data has been added as Fig 1A.

Reviewer: Analysis of a further mouse model with a clearer phenotype would greatly improve the paper.

Authors: The authors agree in that regard that including many mouse models always makes results stronger. However, we strongly disagree with the reviewer that a severe phenotype would be more relevant to assess a therapy effect. To our opinion, to get relevant information of a disease one has to create and use mouse models that replicate findings of a disease, preferably ones carrying patient mutations. These are rare, but our Deletor mice are such a model for adult-onset mitochondrial myopathy; they carry a patient mutation, replicate the patients' finding of development of oxphos-deficient muscle fibers and neurons, and accumulate multiple mtDNA deletions, exactly as our patients with the same disease and mutation. The key physiological findings in these mice have been replicated in molecular level in patients, even providing new biomarkers for diagnostics (PNAS 2005; Hum Mol Genet 2010; Lancet Neurology 2011). Most mitochondrial mouse models used in the field are tissue-specific knock-outs, because full knock-outs are embryonic lethal. We strongly argue that the relevance of such models to real disease physiology is restricted. 1) the target cell type is selected, and therefore contribution of organismal responses, or even neighboring cells, cannot be considered. 2) Knock-out of an essential mitochondrial protein leads always to death. The narrow window of dysfunction before cell death, and potentially extended lifespan, are used to evaluate a therapy effect. Those models may be relevant for early-onset disorders of childhood, but not for late-onset disorders. The special strength of our study is the very close resemblance of the physiological changes of the disease to human findings, and subtle progressive deterioration of mitochondrial function without restriction of lifespan. This allows long-term treatment and assessment of disease progression without lethality as an end-point. Good disease models provide a solid basis for progressing to human studies and direct an optimal patient group to be selected: in this case, patients with mitochondrial myopathies.

Referee #3 (Remarks): The Authors have previously shown the presence of a pseudo-starvation response in their model, the Deletor mice, implying involvement of abnormal nutrient signaling in this disease model. In the present work, they hypothesize that respiratory chain dysfunction, by reducing NADH utilization and thus the NAD⁺: NADH⁺ ratio attenuates mitochondrial biogenesis mediated by the nutrient sensor Sirt1.

Since induction of mitochondrial biogenesis could be beneficial in the treatment of respiratory chain disease, the authors modify the supposed NAD/NADH⁺ imbalance due to an inefficient respiratory chain function in their Deletor model, by administering NAD⁺ to the animals. In fact NAD⁺ has been previously shown by Canto & et al in 2012 to promote mitochondrial biogenesis and oxidative metabolism in ageing mice by increasing the NAD⁺/NADH ratio.

Authors: We agree with these comments, and note that Canto et al. 2012 publication indeed was the basis of this study, and we replicate their study protocol in detail. This work has also been referenced in our manuscript accordingly, but now stated even more clearly.

Reviewer: The results of the study nicely confirm the previous observations by Canto & et al.

Authors: Yes, we completely agree.

Reviewer: Furthermore they provide evidence for a positive effect of NR on the mitochondrial unfolded protein response.

Authors: Yes, and this is a) the first indication of classical unfolded protein response in a mitochondrial disease model and b) the first indication of UPR_{mt} induction after NAD⁺ precursor supplementation.

Reviewer: Based on their observations, the Authors propose that chronic NR supplementation, by inducing mitochondrial biogenesis, slows down the progression of mitochondrial myopathy in the Deletor mice, and suggest this therapeutic strategy for patients with late-onset mitochondrial myopathy. My major concern regards the demonstration of a clear link between RCD and decreased NAD⁺ in their mice. In other words: while the Authors clearly demonstrate that chronic NR administration increases mitochondrial biogenesis, respiratory chain complexes proteins, and histochemical COX activity to the same extent in the Del and in control mice, they do not provide enough evidence linking lower levels of NAD⁺ to RCD in the skeletal muscle of Del mice, before NR treatment. In fact, they provide information about NAD⁺ levels only in liver (as stated in the text, results section) and these are not significantly different from controls.

Authors: Please see the comments above on liver NAD⁺ measurements to the same Reviewer and the Editor. We would like to emphasize that we have now set up new methodology based on mass spectrometry and detect decreased NAD⁺ in old Deletors. Our findings and previous literature strongly support NR to increase NAD⁺ in mouse tissues, which has obvious beneficial effects for MM mice.

Reviewer: In addition, according to the legend of figure 4A, where NAD⁺ values are reported, evaluations were performed on liver and muscle. If this was the case, they are not kept distinct in the graph, which is confusing.

Authors: Thank you for noting this error. We showed originally only liver NAD⁺ level determination, not muscle. New data on the subject, as explained in detail above, have been included to new Fig 1A, and Suppl Fig 1.

Reviewer: Moreover, since there is no clear evidence of a defective respiratory chain in the Del mouse, that only shows morphologic (histochemical and ultra structural) features consistent with mitochondrial myopathy, we are aware that could be hard to demonstrate a NAD⁺ reduction.

Authors: The diagnostic hallmark and the direct indication of a respiratory chain deficiency in human patients with adult-onset mitochondrial myopathies are indeed the histochemical findings of reduced activity of cytochrome c oxidase and consequent increased activity of succinate dehydrogenase, *in situ*, in muscle fibers. This same finding is a key phenotype of the Deletor mice with mitochondrial myopathy. COX-deficiency (i.e. respiratory chain deficiency) is clearly demonstrated by COX histochemical assay, which we performed (Fig 1B, C). Often “COX-negative fibers” are mistaken as negative for protein, i.e. immunodetection, which is not the case here – we measure *in situ* activity on frozen sections of muscle. Therefore, the mice clearly have respiratory chain deficiency.

We do agree with the reviewer that because of the mild phenotype, changes in steady-state NAD⁺ may be hard to demonstrate. Despite that, our NAD⁺ measurements show the trend of increasing levels after NR treatment, completely consistent to findings of Canto et al in WT mice.

Rev. To confirm the Author's hypothesis, it would be important to use a mouse model with a more clear phenotype of mitochondrial myopathy, possibly also with evidence of impaired exercise capacity.

Authors: Please see the comment above, for this same reviewer. We have very solid findings of NR effect on mitochondrial morphology, ultrastructure and enzyme activities, as well as molecular responses including UPRmt.

Reviewer: A second issue regards the analysis of other tissues besides skeletal muscle: it is always good to provide additional data that may help in understanding the mechanisms at play. However, sometimes it is difficult to understand from the text and the figures which results were obtained in which tissue, as is the case for the paragraph on steady state NAD⁺ amount and FOXO deacetylation and the relative figures 4 A-F. The results are mixed up in the text and figures and should be kept separated, since the main scope of this work is to demonstrate that NAD⁺ is effective in slowing down the progression of mitochondrial myopathy.

Authors. Thank you for the comment. We have now clarified further in Figure 4 legends, to indicate in detail which was the tissue of origin in each analysis. In other figures the tissues are clearly marked in each figure legend.

Reviewer: A third issue concerns the finding of significantly lower mutation load in the muscle of Del mice receiving NAD⁺. Could the Authors comment on the link between the induction of mitochondrial biogenesis by NAD⁺ and the decreased amount of mtDNA deletion in muscle?

Au: Please see our comment on the subject for Rev 2. We have added a sentence on this to the discussion.

2nd Editorial Decision

10 March 2014

Thank you for the submission of your revised manuscript to EMBO Molecular Medicine. We have now received the enclosed reports from the Reviewers that were asked to re-assess it. As you will see the Reviewers are now globally supportive and I am pleased to inform you that we will be able to accept your manuscript pending the following final amendments:

- 1) Reviewer 3 is still concerned that the evidence in support for your closing remark "Our data show that conflicting nutrient signaling is integral for MM pathogenesis and disease progression" is not sufficient. I agree with him/her that the best course of action would be to tone down the statement.
- 2) Please provide better image files for Fig. 1 panel D and Fig. 4 panels A and G. Unfortunately, the resolution of these images seems rather low. This is very apparent, for example, when zooming in especially for Fig. 1D. Also, the boxes defining the individual blots have heterogeneous sizes and thicknesses. These issues will lead to problems when the production team tries to resize these images for the final manuscript.
- 3) As per our Author Guidelines, the description of all reported data that includes statistical testing must state the name of the statistical test used to generate error bars and P values, the number (n) of independent experiments underlying each data point (not replicate measures of one sample), and the actual P value for each test (not merely 'significant' or 'P < 0.05').
- 4) Please provide a short list (up to 5) of bullet points that summarize the key NEW findings in a separate file. The bullet points should be designed to be complementary to the abstract and will be used online in our new web platform.
- 5) We are now encouraging the publication of source data, particularly for electrophoretic gels and blots, with the aim of making primary data more accessible and transparent to the reader. Would you be willing to provide a PDF file per figure that contains the original, uncropped and unprocessed scans of all or at least the key gels used in the manuscript? The PDF files should be labeled with the appropriate figure/panel number, and should have molecular weight markers; further annotation may be useful but is not essential. The PDF files will be published online with the article as supplementary "Source Data" files. If you have any questions regarding this just contact me.

I look forward to reading a new revised version of your manuscript as soon as possible. Obviously, the sooner we receive it, the sooner we will be able to accept for publication.

***** Reviewer's comments *****

Referee #1 (Remarks):

My concerns were substantially satisfied

Referee #2 (Remarks):

The figure is now much more convincing and all my comments have been adequately addressed.

Referee #3 (Comments on Novelty/Model System):

In the revised manuscript, the results are more clearly detailed in figure 4. By adding the analysis of NR+ levels in skeletal muscle of Deletor mice, the authors do not provide enough evidence for decreased NR+ tissue levels, even at a very old age. However, they provide robust evidence linking NR administration to mt biogenesis etc, both in terms of effect and underlying mechanisms. This support their conclusion that "....fine-tuning of cellular signaling with vitamin cofactors, especially NAD+ precursors, are an intriguing and straightforward therapeutic strategy that should be explored in patients with late-onset mitochondrial myopathy, the most common type of adult-onset mitochondrial disorders".

I still have concerns about their first conclusive statement: "our data show that conflicting nutrient signaling is integral for MM pathogenesis and disease progression". To me there is not enough evidence provided in support of this statement. Probably the best thing would be to change the word "show" with the word "suggest".

Referee #3 (Remarks):

I have no major concerns about the revised manuscript

2nd Revision - authors' response

14 March 2014

Thank you for the submission of your revised manuscript to EMBO Molecular Medicine. We have now received the enclosed reports from the Reviewers that were asked to re-assess it. As you will see the Reviewers are now globally supportive and I am pleased to inform you that we will be able to accept your manuscript pending the following final amendments:

1) Reviewer 3 is still concerned that the evidence in support for your closing remark "Our data show that conflicting nutrient signaling is integral for MM pathogenesis and disease progression" is not sufficient. I agree with him/her that the best course of action would be to tone down the statement.

Au: We have now revised the sentence, as suggested by the reviewer 3: “Our data **suggest** that conflicting nutrient signaling is integral for MM pathogenesis and disease progression.”

2) Please provide better image files for Fig. 1 panel D and Fig. 4 panels A and G. Unfortunately, the resolution of these images seems rather low. This is very apparent, for example, when zooming in especially for Fig. 1D. Also, the boxes defining the individual blots have heterogeneous sizes and thicknesses. These issues will lead to problems when the production team tries to resize these images for the final manuscript.

Au: We have now confirmed that all Western blot figures are of 1200 dpi.

3) As per our Author Guidelines, the description of all reported data that includes statistical testing must state the name of the statistical test used to generate error bars and P values, the number (n) of independent experiments underlying each data point (not replicate measures of one sample), and the actual P value for each test (not merely 'significant' or 'P < 0.05').

Au: We have now added exact p-values and Statistical tests, as well as indicated independent experiments in the figure legends. In Fig 3A-D, the p-values could not be fitted to the figure.

4) Please provide a short list (up to 5) of bullet points that summarize the key NEW findings in a separate file. The bullet points should be designed to be complementary to the abstract and will be used online in our new web platform.

Au: We have now included these bullet points as a separate file:

- Nicotinamide riboside, vitamin B3, delays progression of mitochondrial myopathy
- Nicotinamide riboside cures pathology-associated mitochondrial ultrastructure
- Nicotinamide riboside improves mitochondrial DNA stability
- Mitochondrial disease induces mitochondrial unfolded protein response, further enhanced by nicotinamide riboside
- Nicotinamide riboside is a promising treatment for adult-onset mitochondrial myopathy

5) We are now encouraging the publication of source data, particularly for electrophoretic gels and blots, with the aim of making primary data more accessible and transparent to the reader. Would you be willing to provide a PDF file per figure that contains the original, uncropped and unprocessed scans of all or at least the key gels used in the manuscript? The PDF files should be labeled with the appropriate figure/panel number, and should have molecular weight markers; further annotation may be useful but is not essential. The PDF files will be published online with the article as supplementary "Source Data" files. If you have any questions regarding this just contact me.

Au: We have now provided Source Data for all key results, labeled as requested.

***** Reviewer's comments *****

Referee #1 (Remarks):

My concerns were substantially satisfied

Au: OK.

Referee #2 (Remarks):

The figure is now much more convincing and all my comments have been adequately addressed.

Au: OK.

Referee #3 (Comments on Novelty/Model System):

In the revised manuscript, the results are more clearly detailed in figure 4. By adding the analysis of NR+ levels in skeletal muscle of Deletor mice, the authors do not provide enough evidence for decreased NR+ tissue levels, even at a very old age. However, they provide robust evidence linking NR administration to mt biogenesis etc, both in terms of effect and underlying mechanisms. This support their conclusion that "...fine-tuning of cellular signaling with vitamin cofactors, especially NAD+ precursors, are an intriguing and straightforward therapeutic strategy that should be explored in patients with late-onset mitochondrial myopathy, the most common type of adult-onset mitochondrial disorders".

I still have concerns about their first conclusive statement: "our data show that conflicting nutrient signaling is integral for MM pathogenesis and disease progression". To me there is not enough evidence provided in support of this statement. Probably the best thing would be to change the word "show" with the word "suggest".

Au: corrected as suggested by the reviewer.

Referee #3 (Remarks):

I have no major concerns about the revised manuscript

Au: ok.